# The Profile of Bicycle Users, Their Perceived Difficulty to Cycle, and the Most Frequent Trip Origins and Destinations in Aracaju, Brazil

**DOI:** 10.3390/ijerph17217983

**Published:** 2020-10-30

**Authors:** Mabliny Thuany, João Carlos N. Melo, João Pedro B. Tavares, Filipe M. J. Santos, Ellen C. M. Silva, André O. Werneck, Sayuri Dantas, Gerson Ferrari, Thiago H. Sá, Danilo R. Silva

**Affiliations:** 1Department of Physical Education, Federal University of Sergipe (UFS), São Cristóvão 49100-000, Brazil; mablinysantos@gmail.com (M.T.); joaofghc@gmail.com (J.C.N.M.); jp.edf1@gmail.com (J.P.B.T.); filipe.matheusef@gmail.com (F.M.J.S.); ellencmendesilva@gmail.com (E.C.M.S.); danilorpsilva@gmail.com (D.R.S.); 2Center for Epidemiological Research in Nutrition and Health (NUPENS), University of São Paulo (USP), São Paulo 01246-904, Brazil; andreowerneck@gmail.com (A.O.W.); thiagoherickdesa@gmail.com (T.H.S.); 3Non-Governmental Organization Associação Ciclo Urbano, Aracaju 49070-376, Brazil; sayuriods@gmail.com; 4Physical Activity, Sport and Health Sciences Laboratory, Faculty of Medical Sciences, University of Santiago, Chile (USACH), Santiago 7500618, Chile

**Keywords:** transportation, exercise, sedentary behavior, urbanization, city planning

## Abstract

The objective of this study was to describe the profile of bicycle users, their perceived difficulty to cycle, and the most frequent trip origins and destinations in Aracaju, Northeast Brazil. Our cross-sectional study sampled 1001 participants and we collected information through structured interviews. Aged ≥15 years, participants were residents of all Aracaju’s neighborhoods and used a bicycle for commuting to work or for leisure. We observed that bicycle users in Aracaju are predominantly employed male subjects, aged between 18 and 40 years, and were the heads of their households. Most of the them reported “work” as the main reason for their bicycle trips and, “health” and “practicality” aspects as their main motivations for using bicycles. In general, the neighborhoods in the north and center of the city were identified as the most difficult for cycling, and the easiest trips occurred in places with cycle paths. As a conclusion of this study, we reaffirm the need for intersectoral actions that create favorable environments for active commuting and more sustainable cities.

## 1. Introduction

Increased physical inactivity is currently a major challenge in several countries [1], mainly due to its association with an increased risk of chronic noncommunicable diseases, such as cardiovascular diseases, different types of cancer, and mental disorders [2,3]. Most people do not comply with the minimum recommended levels of physical activity to prevent diseases and to protect health, i.e., according to the World Health Organization’s physical activity guidelines [4]. In this sense, the adoption of different strategies to provide an active lifestyle emerges as the most viable alternative for the promotion of physical activity [1,5], such as changes in the urbanized environment of cities to enable active commuting [6].

Given that commuting is seen as a daily practice among adult populations and that at least 30 min a day are spent on this activity [7], active alternatives, such as the use of bicycles, can be applied as a strategy to increase total physical activity. Commuting by bicycle provides additional benefits, such as contributing to air and noise pollution reduction, increasing social engagement, and reducing road traffic injuries [8,9,10,11,12]. Active commuting by bicycle can be considered as an accessible and relatively “easy” way to promote physical activity among different age groups (children, adults, and older adults) [13,14,15,16] and cultures/countries [17,18]. However, despite the relevance of this approach, there is limited information about the effectiveness of these strategies or even the adherence of the population to them, particularly in low- and middle-income settings [19,20,21].

Previous studies report the influence of individual and environmental variables on the use of bicycles for transport, including sex, age, economic status, and education, in addition to proximity between the point of origin and the destination [19,22,23,24]. In the Brazilian context, the accessibility of bicycle use as a means of transport is a topic of discrepancy. Reis, et al. [25] compared the prevalence of bicycle use for transportation among three cities in different states and regions of Brazil, observing differences between Recife (Pernambuco, Northeast, 16.0%), Curitiba (Paraná, South, 9.6%), and Vitória (Espírito Santo, Southeast, 8.8%). In another study conducted in Rio Claro (São Paulo, Southeast), Teixeira, et al. [26] found a much greater prevalence of bicycle use for transportation (28.3%). Although it seems clear that men, younger adults, and lower education/economic status were associated with greater use of bicycles for transportation [8,25,26,27] in both studies, some specificities should be considered in the low- and middle-income contexts. For example, Reis, et al. [25] observed a higher prevalence of bicycle use in the city with the highest crime rate (Recife), which was not expected. However, this was also the city with the lowest human development index, highest unemployment rate, and social inequalities, suggesting that bicycle use could not be an option in low-income regions. Thus, understanding the profile of bicycle users and their relations with specific characteristics of the cities is justified in order to provide better conditions for those who already use the bicycle, and to create opportunities for other population subgroups to use bicycles for transport.

Aracaju city (Sergipe, Northeast) is known as of the first Brazilian capitals to implement the proposal for mobility on bicycles in 2005 (implementing networks of cycle paths and on flat land). However, only 11.9% of the population reported active commuting (walking and cycling) in 2018 [28], and there is no available information on the use of the bicycle for transport. Given that Aracaju’s street design favors the use of bicycles for commuting, studies on bicycle flow within available structures would contribute to the development of strategies for urban planning, acting as an important way to increase health, environment, and sustainability indicators in that context. However, information about bicycle users, their perceptions about the city, and the most frequent trip origins and destinations could foster public policies in urban planning. Thus, we researched the profile of bicycle users, their perceived difficulty to cycle, and the most frequent origins and destinations of bicycle travel in Aracaju, Brazil.

## 2. Methods

### 2.1. Design

We used a cross-sectional method, carried out in Aracaju, Brazil. Aracaju is the state capital of Sergipe, with about 657 thousand inhabitants (2019), and a Human Development Index of 0.770 (2010). The population’s average income in 2010 was 3.1 times the minimum wage (approximately USD 200) and 56.6% of public spaces were forested [29].

### 2.2. Sample and Data Procedures

The information in this study was obtained from the “Origin and Destination of Biking Trips in the City of Aracaju, Survey,” over a one-year period (June 2014 to June 2015). The survey was conducted with bicycle users that were approached personally in all 40 neighborhoods of the city, respecting the proportionality of the population of each neighborhood [29]. The structured interviews were conducted by 17 trained advisors on weekdays, from 2:00 PM to 7:00 PM. This convenience sample consisted of 1001 bicycle users. The city neighborhoods were organized into bordering zones (north, south, central, and expansion). It is important to note that the set of neighborhoods, communities, and villages that make up the Expansion Zone have been incorporated to Aracaju by legal decision. All participants were provide with information about the objectives of the study, which was conducted in accordance with the ethical standards of the institutional and/or national research committee, respecting the 1964 Helsinki Declaration and its further amendments, or comparable ethical standards. The study was approved by the Ethics Committee of the Federal University of Sergipe (CAAE: 16418619.7.0000.5546).

### 2.3. Instrument

The questionnaire used was prepared by the nongovernmental organization “Associação Ciclo Urbano—Aracaju,” consisting of 23 items and divided into (a) cyclist profile: sex (male and female), age (categorized into five age groups: up to 18 years, 18 to 30 years, 30 to 40 years, 40 to 50 years, and over 50 years), and family role (head of family, spouse, child, or relative); (b) socioeconomic information: work status (employed and not employed), educational level (below upper secondary,, secondary, and above secondary), activity sector (commerce, industry, construction, education, and health), monthly income (up to one minimum wage and above one minimum wage), and automotive vehicle ownership (yes or no); (c) characteristics of the origin and destination of trips made by bicycles: reason for the trip (work, school, leisure, shopping, and others), location of origin and destination (Aracaju neighborhood), region’s access conditions (easy, difficult), bicycle parking conditions (public, paid parking, and free parking), motivation to ride a bicycle (health, practicality, leisure, economic, and two options), departure and arrival period of the day (morning, afternoon, and night), time spent commuting (0 to 15 min, 15 to 30 min, 30 to 45 min, 45 to 60 min, and over 60 min), whether the destination is a different neighborhood (yes or no), and the type of trip origin and destination (nonrecreational or recreational).

### 2.4. Statistical Analysis

Descriptive information was presented using absolute and relative frequencies. To compare the profile of bicycle users with Aracaju’s population, we restricted the analysis to adult participants and used the information provided by the Brazilian 2013 National Health Survey, which is the closest survey to the reference year with representative information of the adult population of Aracaju [30] (% and 95% confidence intervals). The perception analysis of cycling difficulty in neighborhoods was determined by the relative frequency of citations used (easy or difficult). Absolute frequencies of commuting between neighborhoods were used to verify the main trip origin and destination, and the main neighborhoods cited as the origin or destination. Main trip origins and destinations were defined as those representing at least 0.9% of the total mentioned. The Aracaju Mobility Master Plan (2016) was used to identify bike lanes that were implemented until 2015. Finally, the information was presented in the form of a map with an origin and destination analysis. It was carried out using a cross-reference table to check the main neighborhoods in which there was a greater flow of bicycle users entering and leaving, in addition to identifying the main trip origin and destination of the interviewees. The maps were built using the CorelDRAW Graphics Suite 2019 software (Corel Corporation, Ottawa, ON, Canada). All analyses were performed using SPSS 22.0 software (IBM, Armonk, NY, USA).

## 3. Results

Table 1 shows information on bicycle user profiles. It shows that the sample was predominantly composed of men, aged 18 to 40 years, who were heads of their households and were employed. Most reported schooling at primary and secondary levels, and work in civil construction and health sectors; 66.7% reported “work” as the main reason for their travel. Most individuals reported not having their own automotive vehicle and having income of up to one minimum wage. Most participants reported “healthy” and “practicality” as their main motivation to ride a bicycle.

Figure 1 presents the perceived degree of difficulty for cycling in city neighborhoods. In general, Center (23%) and Porto Dantas (13%) were pointed out as neighborhoods in which cycling was the most difficult for cycling, while 13 de Julho (14%), Atalaia (13%), Siqueira Campos (13%), and Jabotiana (12%) were cited as the ones in which cycling was the easiest.

Figure 2 shows the most frequent trip origins and destinations among the participants. The main destination was Santa Maria neighborhood (10.9%), followed by Atalaia (5.3%), Farolândia (5.7%), and Santos Dumont (6%). Regarding the trip origin and destination, Center (6.6%), Farolândia (5.9%), and Siqueira Campos (5.6%) were the most reported. In addition, there is a greater tendency to move toward neighborhoods close to the point of departure, with 23% of the total trips occurring within the neighborhood of origin (especially the José Conrado (50%) and the Expansion Zone (77%) neighborhoods).

## 4. Discussion

This study aimed to describe the profile of bicycle users, their perceived difficulty to cycle, and the most frequent trip origins and destinations in the city of Aracaju. The results indicated that (1) the 60-km cycle paths distributed in the city of Aracaju serve mostly men, younger adults, and people with lower educational levels, as compared with the population of Aracaju; (2) the use of active commuting is associated with going to work, especially in the lowest income group; (3) most bicycle users move from central to peripheral areas; and (4) the majority of the participants spent an average of 15 to 30 min (per cycling trip). This information is vital to develop strategies to improve current bicycle user conditions and to create new opportunities for less represented population groups, especially in developing countries.

Bicycle user profiles differed greatly from those of Aracaju’s population in general, which reinforces the fact that some population subgroups are more inclined to use bicycles for transportation. The initial results confirm data from previous studies in Brazilian cities showing a similar bicycle user profile, including a higher proportion of adult young men [8,25,26,27]. This behavior in men is associated with a duality in terms of stimuli directed at boys and girls in childhood, with a tendency to maintain these habits in adulthood [31], culminating in less use of bicycles by women in particular and less engagement in physical activities in general. Another factor is the perception of safety, which tends to be different between sexes, making factors related to “lack of lighting during commuting,” “driver-cyclist behavior in traffic,” and “public insecurity” great obstacles, especially among women [16,32]. Some studies [8,26] also showed greater use of bicycles among workers in areas with a higher representation of men (e.g., construction and industry).

Considering that most participants who traveled by bicycle did not own automotive vehicles—a proportion higher than that observed in the population of Aracaju (78.4% vs. 49.0%)—the socioeconomic structure of the population may be a factor that explains these results. In previous studies, this factor has already been negatively associated with levels of active transport [33] and were observed more frequently in low-income regions. Bicycles tend to be a “cheaper” mode of transport due to inaccessibility of a private vehicle, high fares, and lack of quality in public transport.

Safety issues in public transport and the lack of adequate infrastructure for bicycle use tend to “disable” environments that favor walking or cycling as means to travel to work [34]. Our results indicated that, in general, the districts of the North and Central Zones were identified as the most difficult to cycle in, which may be associated with the fact that most of the Aracaju cycle system is concentrated in the South Zone [35]. In previous studies, not having bike paths, conservation of streets and avenues, car traffic, and rough surfaces were examples of variables that could impact bicycle users’ perception of neighborhoods [36,37].

The origin and destination of the trips indicated greater commuting from central areas to peripheral areas of the city. One of the possible explanations may be associated with the growth in investments in civil construction, industry, and commerce in the South Zone of the city, leading people to move to the peripheral regions of the city for work. In addition, a large part of the participants reported working in these sectors (33.9%, 20.7%, and 10.0%, for construction, commerce, and industry, respectively). Another explanation for these results could be the data collection times, which were concentrated in the afternoon and may represent the commute home from work. It was noted that the most frequently cited trip origins and destinations by respondents (Figure 2) involved commuting between adjacent or nearby neighborhoods. Previous studies identified that distances under 10 kilometers were more feasible to retain the use of the bicycle as a means of daily commuting [38,39,40]. In the present study, the most participants spent an average of 15 to 30 min (per trip) on daily commuting. This behavior shows a viability threshold for this mode of transport and can offer health benefits [41]. Other actions could improve and facilitate the use of cycle paths. The equitable distribution of schools, jobs, and sectors necessary for day-to-day activities can help people from places with longest trip origins and destinations [42,43].

The practical application of the study is to use this data as an aid in producing public policies to improve the infrastructure of cycle paths and expand cycle routes. Understanding the behavior and profile of bicycle users may also inform political decisions regarding active transport and urban planning. Although this was a nonrepresentative sample and interviews are likely to provide report bias, this study analyzes the profile and characteristics of trip origins and destinations and the perception of bicycle users in a specific social, economic, and climate context regarding active commuting.

## 5. Conclusions

Bicycle users are predominantly represented by men, aged between 18 and 40 years, from low income families in Aracaju, Brazil. In general, the use of active commuting is associated with going to work, mainly because this form of transport is more practical and healthier than public and private transport. The study reported a tendency of travel from central to peripheral areas, which may be associated with the workplaces. Considering that some bicycle users report the economic factor as a motivation to use bicycles and that women and adolescents are underrepresented in this scenario, the present study reaffirms the need for intersectoral actions to enable the construction of a safer city through expansion of bicycle networks with safe dimensions and accessibility.

## Figures and Tables

**Figure 1 ijerph-17-07983-f001:**
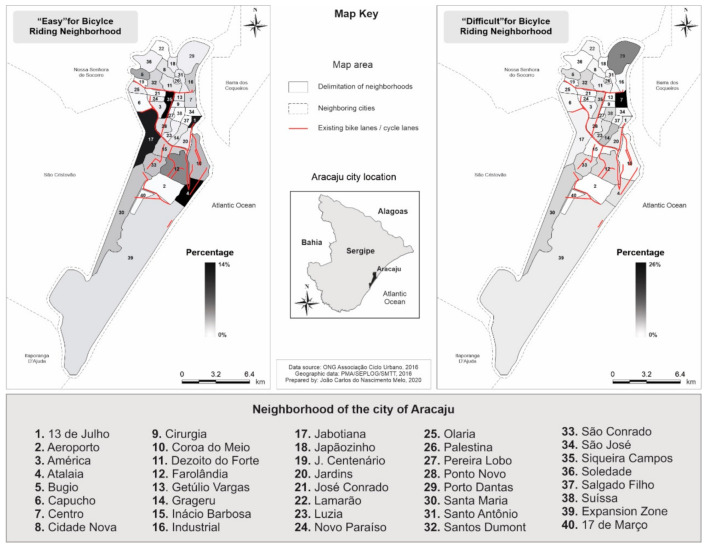
Perception of the difficulty of cycling in Aracaju/Sergipe.

**Figure 2 ijerph-17-07983-f002:**
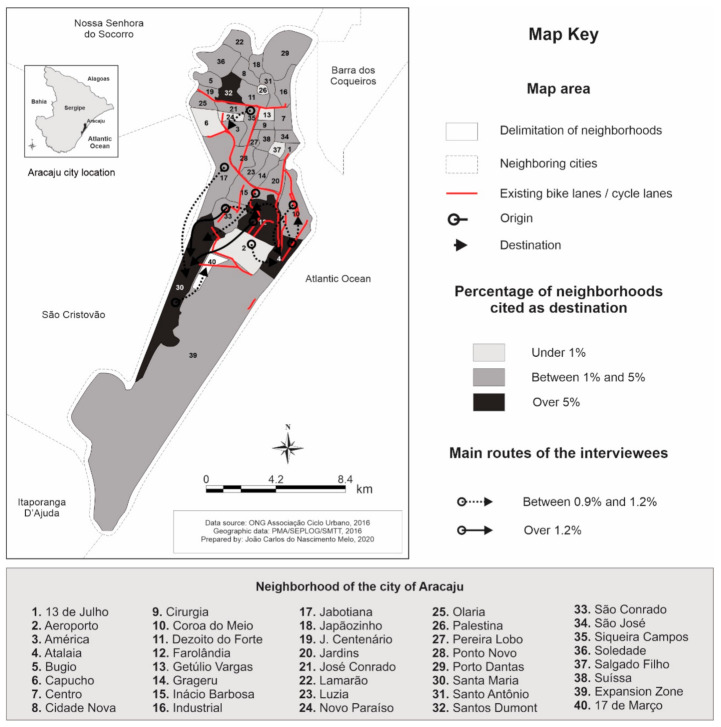
Main Trajectories of Bike Riders in Aracaju/Sergipe.

**Table 1 ijerph-17-07983-t001:** Characteristics of Bicycle Users and Aracaju’s Population.

Variable	Bicycle Users % (*n*)	Population * % (CI 95%)
Sex		
Female	11.5 (102)	55.1 (50.0 to 60.1)
Male	88.5 (782)	44.9 (39.9 to 50.0)
Age		
18–30	34.5 (305)	26.1 (21.8 to 30.8)
30–40	30.8 (272)	24.1 (19.9 to 28.7)
40–50	20.5 (181)	19.2 (15.6 to 23.4)
>50	14.3 (126)	30.7 (26.3 to 35.4)
Family Role		
Head of Family	60.9 (538)	-
Spouse	14.0 (124)	-
Child	19.5 (172)	-
Relative	5.7 (50)	-
Work status		
Employed	88.9 (786)	54.2 (49.3 to 59.2)
Not employed	11.1 (98)	45.7 (40.8 to 50.7)
Education		
Below Secondary	54.3 (480)	38.9 (34.1 to 43.8)
Secondary	32.0 (285)	39.3 (34.5 to 44.2)
Beyond Secondary	13.7 (121)	21.9 (17.9 to 26.5)
Activity Sector		
Commerce	20.7 (179)	-
Industry	10.0 (79)	-
Construction	33.9 (268)	-
Education	2.8 (22)	-
Health	30.6 (242)	-
Monthly Income		
Up to one minimum wage	78.7 (695)	-
Above one minimum wage	21.3 (189)	-
Automotive Vehicle Ownership		
No	78.4 (693)	49.0 (44.0 to 54.1)
Yes	21.6 (191)	51.0 (45.9 to 56.0)
Reason for Trip		
Work	66.7 (590)	-
School	2.9 (26)	-
Leisure	12.7 (112)	-
Shopping	6.4 (57)	-
Others	11.2 (99)	-
Motivation to Ride a Bicycle		
Health	26.4 (229)	-
Practicality	25.3 (219)	-
Leisure	7.5 (65)	-
Economic	19.3 (167)	-
Two options	21.5 (186)	-
Time Spent Commuting		
0 to 15 min	22.7 (200)	-
15 to 30 min	36.4 (321)	-
30 to 45 min	18.1 (160)	-
45 to 60 min	13.9 (123)	-
>60 min	8.8 (78)	-
Is your destination a different neighborhood?		
Yes	82.6 (730)	-
No	17.4 (154)	-
Type of Trip Origin and Destination		
Nonrecreational	96.6 (854)	-
Recreational	3.4 (30)	-

Note: Time spent commuting refers to the total time spent commuting from the place of departure to destination. CI = confidence interval. * Based on the Brazilian 2013 National Health Survey.

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
