# Peer review of "The Profile of Bicycle Users, Their Perceived Difficulty to Cycle, and the Most Frequent Trip Origins and Destinations in Aracaju, Brazil"

_ijerph, 2020, doi:10.3390/ijerph17217983_

Round 1

Reviewer 1 Report

The sampling method for the survey data was not clear, and there is a concern that the data might be biased as too many respondents (87.9%) were male. Were they representative for the whole bicycle users in Aracaju?

Author Response

July, 30rd 2020

Dr. Adilson Marques

Guest Editor, Special Issue "Active Commuting and Active Transportation"

International Journal of Environmental Research and Public Health (IJERPH)

We would like to thank the editor for the opportunity to resubmit the attached manuscript (ijerph-866782) entitled “The profile of bicycle users, their perceived difficulty to cycle and the most used routes in Aracaju – Brazil” for possible publication in the IJERPH.

The authors also wish to express their gratitude for the efforts of the reviewers in directing the manuscript towards a more acceptable form for publication. The manuscript has been carefully checked, and appropriate changes made in accordance with the reviewers’ suggestions. We have highlighted all the revisions in the manuscript using track changes and have attached a separate document (below) containing point-by point responses to the reviewers’ comments.

The authors hope that the added revisions adequately address the comments.

Sincerely,

The authors.

Reviewer 1

"The sampling method for the survey data was not clear, and there is a concern that the data might be biased as too many respondents (87.9%) were male. Were they representative for the whole bicycle users in Aracaju?"

Authors’ answer: The fact that the most participants were male is due to the characteristics of bicycle users in Aracaju. Similar results were also observed in other Brazilian cities (Teixeira et al. 2013; Reis et al. 2013, Kienteka et al. 2014, Sá et al. 2016). In order to strengthen this point, we included information about citywide information available at Table 1 to allow a clear view of the differences between bicycle users and the whole population. We also included more details about the sampling methods.

“The survey was conducted with bicycle users that were approached personally in all 40 neighborhoods of the city, respecting the proportionality of the population of each neighborhood [29]. The structured interviews were conducted by 17 trained assessors during the period of 2:00 pm to 7:00 pm only on weekdays. This convenience sample consisted of 1,001 bicycle users.”

  1. IBGE. Sergipe. Availabe online: https://cidades.ibge.gov.br/brasil/se/aracaju/panorama (accessed on 25/04/2020).

Teixeira, I.; Nakamura, P.; Smirmaul, B.; Fernandes, R.; Kokubun, E. Fatores associados ao uso de bicicleta como meio de transporte em uma cidade de médio porte. Revista Brasileira de Atividade Física & Saúde 2013, 18, 698, doi:10.12820/rbafs.v.18n6p698.

Reis, R.S.; Hino, A.A.; Parra, D.C.; Hallal, P.C.; Brownson, R.C. Bicycling and walking for transportation in three Brazilian cities. American journal of preventive medicine 2013, 44, e9-17, doi:10.1016/j.amepre.2012.10.014.

Kienteka, M.; Reis, R.S.; Rech, C.R. Personal and behavioral factors associated with bicycling in adults from Curitiba, Paraná State, Brazil. Cadernos de Saúde Pública 2014, 30, 79-87.

Sa, T.H.; Duran, A.C.; Tainio, M.; Monteiro, C.A.; Woodcock, J. Cycling in Sao Paulo, Brazil (1997-2012): Correlates, time trends and health consequences. Prev Med Rep 2016, 4, 540-545, doi:10.1016/j.pmedr.2016.10.001.

Reviewer 2 Report

My main issue on this paper is its readability and novelty. I have following concerns for the authors to consider. 

1. Although this paper seems to be an empirical study, it didn’t provide any information on the literature review. As a result, I cannot clearly see the contributions of this paper to the research field.

2. The presentation of the methods is rough and needs more details than its current version. The object (including people and geospatial area), the data, and the mathematical approach (if there is) should be given in a more systematic way.

3. I recommend the authors to illustrate their results by using figures or tables, instead of only using language, so that the readers can easily check them.

4. The discussion contains some contents that belongs to the literature review. Also the authors need to number their findings for the sake of readability.

Thanks!

Author Response

July, 30rd 2020

Dr. Adilson Marques

Guest Editor, Special Issue "Active Commuting and Active Transportation"

International Journal of Environmental Research and Public Health (IJERPH)

We would like to thank the editor for the opportunity to resubmit the attached manuscript (ijerph-866782) entitled “The profile of bicycle users, their perceived difficulty to cycle and the most used routes in Aracaju – Brazil” for possible publication in the IJERPH.

The authors also wish to express their gratitude for the efforts of the reviewers in directing the manuscript towards a more acceptable form for publication. The manuscript has been carefully checked, and appropriate changes made in accordance with the reviewers’ suggestions. We have highlighted all the revisions in the manuscript using track changes and have attached a separate document (below) containing point-by point responses to the reviewers’ comments.

The authors hope that the added revisions adequately address the comments.

Sincerely,

The authors.

Reviewer 2

General Comments

My main issue on this paper is its readability and novelty. I have following concerns for the authors to consider.

Comment 1. Although this paper seems to be an empirical study, it didn’t provide any information on the literature review. As a result, I cannot clearly see the contributions of this paper to the research field.

Authors’ answer:  We thank the reviewer for this suggestion. We have included more information about the literature review in order to strengthen the introduction section (3rd paragraph; below).

“Reis, et al. [25], comparing the prevalence of bicycle use for transportation among three cities in different states and regions of Brazil, observed differences between Recife (Pernambuco, Northeast, 16.0%), Curitiba (Paraná, South, 9.6%) and Vitória (Espírito Santo, Southeast, 8.8%). In another study conducted in Rio Claro (São Paulo, Southeast), Teixeira, et al. [26] found a much greater prevalence of bicycle use for transportation (28.3%). Taken together, although seems clear that men, younger adults and lower education/economic status were associated to greater use of bicycle for transportation [8,25-27], some specificities should be considered in low and middle-income context. For example, Reis, et al. [25] observed a higher prevalence of bicycle use in the city with the highest crime rate (Recife), which was not expected. However, this was also the city with the lowest human development index, highest unemployment rate and social inequalities, suggesting that bicycle use could not be an option in low-income regions. Thus, understanding the profile of the bicycle users and their relationships with the specific characteristics of the cities are warranted in order to provide better conditions for those who already use the bicycle, and to create opportunities for other population subgroups can use bicycle for transportation.”

  1. Sa, T.H.; Duran, A.C.; Tainio, M.; Monteiro, C.A.; Woodcock, J. Cycling in Sao Paulo, Brazil (1997-2012): Correlates, time trends and health consequences. Prev Med Rep 2016, 4, 540-545, doi:10.1016/j.pmedr.2016.10.001.
  2. Reis, R.S.; Hino, A.A.; Parra, D.C.; Hallal, P.C.; Brownson, R.C. Bicycling and walking for transportation in three Brazilian cities. American journal of preventive medicine 2013, 44, e9-17, doi:10.1016/j.amepre.2012.10.014.
  3. Teixeira, I.; Nakamura, P.; Smirmaul, B.; Fernandes, R.; Kokubun, E. Fatores associados ao uso de bicicleta como meio de transporte em uma cidade de médio porte. Revista Brasileira de Atividade Física & Saúde 2013, 18, 698, doi:10.12820/rbafs.v.18n6p698.
  4. Kienteka, M.; Reis, R.S.; Rech, C.R. Personal and behavioral factors associated with bicycling in adults from Curitiba, Paraná State, Brazil. Cadernos de Saúde Pública 2014, 30, 79-87.

Comment 2. The presentation of the methods is rough and needs more details than its current version. The object (including people and geospatial area), the data, and the mathematical approach (if there is) should be given in a more systematic way.

Authors’ answer: As suggested, we have included information throughout the methods section. Specifically, we provide further details about the number of assessors, criteria of neighborhood choice, data collection and analyses.  

Comment 3. I recommend the authors to illustrate their results by using figures or tables, instead of only using language, so that the readers can easily check them.

Authors’ answer: We thank the reviewer for this suggestion. In addition to the text description, one table and two figures were used to present our results. We also added additional information regarding the profile of the population of Aracaju at Table 1.  

Comment 4. The discussion contains some contents that belongs to the literature review. Also the authors need to number their findings for the sake of readability.

Authors’ answer: We have reduced the literature review contents in the discussion section. We also enumerated the main findings of the study.

“The results indicated that: 1) the 60 km cycle paths distributed in the city of Aracaju serve mostly to men, younger adults and people with lower educational levels, as compared with the population of Aracaju; 2) the use of active commuting is associated with going to work, especially in the lowest income group; 3) most bicycle users move from central to the peripheral areas; and that 4) the majority of the participants spent an average of 15 to 30 min (per cycling trip).”

Reviewer 3 Report

The authors present a socio-demographic and (zonal) origin-destination profile of cyclists in a mid-sized Brazilian city based on an intercept survey of cyclists. The cyclists were also asked to self-rate areas of the city as "easy" or "difficult" to cycle. There is certainly increasing interest in cyclist typologies and qualitative assessments of cycling conditions, and this article represents a contribution on those fronts from a region we've heard less about in the literature. However, the current article does not, for me, rise to a sufficient level of scientific contribution to make it worthy of publication. In addition, the use of "route" here is potentially confusing as it refers to an established and active area of research regarding the specific network paths chosen by cyclists, while here it really refers to cycling trips and endpoints but not the route in between. I would encourage the authors to re-think the presentation of results--there may be enough data here to dive deeper into the profile of cyclists and trip descriptions, but as it stands now I don't feel there's enough depth to justify a full paper.

Author Response

July, 30rd 2020

Dr. Adilson Marques

Guest Editor, Special Issue "Active Commuting and Active Transportation"

International Journal of Environmental Research and Public Health (IJERPH)

We would like to thank the editor for the opportunity to resubmit the attached manuscript (ijerph-866782) entitled “The profile of bicycle users, their perceived difficulty to cycle and the most used routes in Aracaju – Brazil” for possible publication in the IJERPH.

The authors also wish to express their gratitude for the efforts of the reviewers in directing the manuscript towards a more acceptable form for publication. The manuscript has been carefully checked, and appropriate changes made in accordance with the reviewers’ suggestions. We have highlighted all the revisions in the manuscript using track changes and have attached a separate document (below) containing point-by point responses to the reviewers’ comments.

The authors hope that the added revisions adequately address the comments.

Sincerely,

The authors.

Reviewer 3

General Comments

The authors present a socio-demographic and (zonal) origin-destination profile of cyclists in a mid-sized Brazilian city based on an intercept survey of cyclists. The cyclists were also asked to self-rate areas of the city as "easy" or "difficult" to cycle. There is certainly increasing interest in cyclist typologies and qualitative assessments of cycling conditions, and this article represents a contribution on those fronts from a region we've heard less about in the literature. However, the current article does not, for me, rise to a sufficient level of scientific contribution to make it worthy of publication.

Authors’ answer: We thank the reviewer for the positive criticism about our study. We reviewed the structure of the paper, adjusting terms and trying to improve their scientific contribution. We hope we could achieve some of your concerns and we keep open to further suggestions. We believe that the paper has its scientific value as it provides novel and detailed information about the profile of bicycle users and their perceived difficulty to cycle in Aracaju, Brazil, an area in which much less information exists in comparison to wealthier areas of the country. Aracaju is coastal city that shares historical, topographic and economic similarities with many other cities from the Northeast region of Brazil, and certainly a better parameter for those places than the cities from the Southeast or South region of the country, where most of the evidence has been produced.

In addition, the use of "route" here is potentially confusing as it refers to an established and active area of research regarding the specific network paths chosen by cyclists, while here it really refers to cycling trips and endpoints but not the route in between.

Authors’ answer: We agree with the reviewer and “routes” was replaced by “trips origin and destination” throughout the text.

I would encourage the authors to re-think the presentation of results--there may be enough data here to dive deeper into the profile of cyclists and trip descriptions, but as it stands now I don't feel there's enough depth to justify a full paper.

Authors’ answer: As suggested by the reviewers, we have added additional information regarding the profile of the population of Aracaju at Table 1. We think this strengthen the interpretation of the results, making clearer the profile of bicycle users against the whole population of the city.

Reviewer 4 Report

It is an interesting topic. This paper mainly describes the profile of bicycle users, their perceived difficulty to cycle, and the most used routes in Aracaju, Northeast Brazil,in order to provide reference information for public policies of urban planning.

The introduction is logical and progressive, and the research problem is clear. The article begins by suggesting that physical inactivity in several countries is a serious threat to health. Then the strategies adopted by countries to promote physical activities are proposed. As commuting is a daily necessity, the use of bicycles as a more active alternative can effectively increase total physical activity. Therefore, it is necessary to consider the factors affecting the acquisition of cycling. Then it introduces the factors that influence the way the population moves in the Brazilian context, the implementation of the proposal for mobility on bicycles in the Brazilian capital, and the importance of urban planning. Finally, the research problem of this paper is extended.

However, there are still some minor problems I think need further improvement. Firstly, The literature review is insufficient, which does not cover the literature related to the topic well, so that readers can’t have a general cognition of the related articles in this field, which is not conducive to a profound understanding of the research value and innovation points of this paper. Secondly, the method used in the paper is structured interviews, mainly focused on descriptive analysis. If the paper could combine the measurement methods, the results may be more convincing.

Taken as a whole, this is a practical paper that can actually provide references for the improvement of public policy and urban planning.

Author Response

July, 30rd 2020

Dr. Adilson Marques

Guest Editor, Special Issue "Active Commuting and Active Transportation"

International Journal of Environmental Research and Public Health (IJERPH)

We would like to thank the editor for the opportunity to resubmit the attached manuscript (ijerph-866782) entitled “The profile of bicycle users, their perceived difficulty to cycle and the most used routes in Aracaju – Brazil” for possible publication in the IJERPH.

The authors also wish to express their gratitude for the efforts of the reviewers in directing the manuscript towards a more acceptable form for publication. The manuscript has been carefully checked, and appropriate changes made in accordance with the reviewers’ suggestions. We have highlighted all the revisions in the manuscript using track changes and have attached a separate document (below) containing point-by point responses to the reviewers’ comments.

The authors hope that the added revisions adequately address the comments.

Sincerely,

The authors.

Reviewer 4

General Comments

It is an interesting topic. This paper mainly describes the profile of bicycle users, their perceived difficulty to cycle, and the most used routes in Aracaju, Northeast Brazil, in order to provide reference information for public policies of urban planning.

Authors’ answer: We thank the reviewer for the positive feedback.

The introduction is logical and progressive, and the research problem is clear. The article begins by suggesting that physical inactivity in several countries is a serious threat to health. Then the strategies adopted by countries to promote physical activities are proposed. As commuting is a daily necessity, the use of bicycles as a more active alternative can effectively increase total physical activity. Therefore, it is necessary to consider the factors affecting the acquisition of cycling. Then it introduces the factors that influence the way the population moves in the Brazilian context, the implementation of the proposal for mobility on bicycles in the Brazilian capital, and the importance of urban planning. Finally, the research problem of this paper is extended.

Authors’ answer: We thank the reviewer for the positive feedback.

However, there are still some minor problems I think need further improvement. Firstly, The literature review is insufficient, which does not cover the literature related to the topic well, so that readers can’t have a general cognition of the related articles in this field, which is not conducive to a profound understanding of the research value and innovation points of this paper. Secondly, the method used in the paper is structured interviews, mainly focused on descriptive analysis. If the paper could combine the measurement methods, the results may be more convincing.

Authors’ answer: As suggested, we included further studies conducted in the Brazilian context in order to allow a broader view about the specific literature and gaps that this study covers (3rd paragraph of the introduction; below). About the second point, data collection was based on structured interviews only and we did not have other measurements methods. However, after this first approach based on secondary data, this research group aims to further understand aspects related to active transportation in Aracaju and other cities in the context of the Brazilian northeast.

“Reis, et al. [25], comparing the prevalence of bicycle use for transportation among three cities in different states and regions of Brazil, observed differences between Recife (Pernambuco, Northeast, 16.0%), Curitiba (Paraná, South, 9.6%) and Vitória (Espírito Santo, Southeast, 8.8%). In another study conducted in Rio Claro (São Paulo, Southeast), Teixeira, et al. [26] found a much greater prevalence of bicycle use for transportation (28.3%). Taken together, although seems clear that men, younger adults and lower education/economic status were associated to greater use of bicycle for transportation [8,25-27], some specificities should be considered in low and middle-income context. For example, Reis, et al. [25] observed a higher prevalence of bicycle use in the city with the highest crime rate (Recife), which was not expected. However, this was also the city with the lowest human development index, highest unemployment rate and social inequalities, suggesting that bicycle use could not be an option in low-income regions. Thus, understanding the profile of the bicycle users and their relationships with the specific characteristics of the cities are warranted in order to provide better conditions for those who already use the bicycle, and to create opportunities for other population subgroups can use bicycle for transportation.”

  1. Sa, T.H.; Duran, A.C.; Tainio, M.; Monteiro, C.A.; Woodcock, J. Cycling in Sao Paulo, Brazil (1997-2012): Correlates, time trends and health consequences. Prev Med Rep 2016, 4, 540-545, doi:10.1016/j.pmedr.2016.10.001.
  2. Reis, R.S.; Hino, A.A.; Parra, D.C.; Hallal, P.C.; Brownson, R.C. Bicycling and walking for transportation in three Brazilian cities. American journal of preventive medicine 2013, 44, e9-17, doi:10.1016/j.amepre.2012.10.014.
  3. Teixeira, I.; Nakamura, P.; Smirmaul, B.; Fernandes, R.; Kokubun, E. Fatores associados ao uso de bicicleta como meio de transporte em uma cidade de médio porte. Revista Brasileira de Atividade Física & Saúde 2013, 18, 698, doi:10.12820/rbafs.v.18n6p698.
  4. Kienteka, M.; Reis, R.S.; Rech, C.R. Personal and behavioral factors associated with bicycling in adults from Curitiba, Paraná State, Brazil. Cadernos de Saúde Pública 2014, 30, 79-87.

Taken as a whole, this is a practical paper that can actually provide references for the improvement of public policy and urban planning.

Authors’ answer: We thank the reviewer for the positive feedback.

Round 2

Reviewer 1 Report

The authors added a detailed explanation about the survey.

Reviewer 2 Report

The revisions are ok. I have no more issues to propose. Thanks.